# Collaboration between Private and Public Genebanks in Conserving and Using Plant Genetic Resources

**DOI:** 10.3390/plants13020247

**Published:** 2024-01-15

**Authors:** Johannes M. M. Engels, Andreas W. Ebert, Theo van Hintum

**Affiliations:** 1Independent Researcher, 06062 Citta’ della Pieve, PG, Italy; 2Independent Researcher, 73529 Schwäbisch Gmünd, Germany; ebert.andreas6@gmail.com; 3Centre for Genetic Resources, The Netherlands (CGN), Wageningen University & Research, 6700 AA Wageningen, The Netherlands; theo.vanhintum@wur.nl

**Keywords:** plant genetic resources, public genebanks, private genebanks, collaboration, conservation, use

## Abstract

Among the most important users of plant genetic resources, conserved predominantly in public genebanks around the world, are public and private plant breeders. Through their breeding efforts, they contribute significantly to global, regional, and local food and nutrition security. Plant breeders need genetic diversity to be able to develop competitive new varieties that are adapted to the changing environmental conditions and suit the needs of consumers. To ensure continued and timely access to the genetic resources that contain the required characteristics and traits, plant breeders established working collections with breeding materials and germplasm for the crops they were breeding. However, with the changing and increasingly more restrictive access conditions, triggered by new global legal instruments like the Convention on Biological Diversity/Nagoya Protocol and the International Treaty, plant breeders started to establish their own genebanks at the turn of the 21st century. This paper analyses the conditions that contributed to this situation as well as the historical ways that plant breeders used to acquire the germplasm they needed. Public genebanks played and continue to play a conducive role in providing genetic resources to users, including private-sector plant breeders. However, also the practices of the germplasm curators to collect and distribute germplasm were affected by the new legal framework that had been developed in global fora. It is against this background that the complementarity and collaboration between public and private sector genebanks have been assessed. Whenever possible, vegetable genetic resources and vegetable private breeding companies have been used to analyze and illustrate such collaboration. The authors look at reported successful examples of collaborative efforts and consider opportunities and approaches under which such collaboration can be established and strengthened to ensure the continued availability of the building blocks for food and nutrition security.

## 1. Introduction

### 1.1. Motivation

Genetic diversity, within and between species, provides the raw material for plant breeders to work with. This diversity has been readily available and without restrictions for a long time, until the 1980s, albeit increasingly threatened by genetic erosion. Plant breeders typically established and maintained working collections of selected materials of a given crop, including their own breeding lines, collected materials, and accessions obtained from public genebanks or (seed) markets elsewhere. With the introduction of a legal framework for access and benefit-sharing (ABS) of this genetic diversity, in fact, of biodiversity at large, access to genetic resources became more restricted or was even, in some cases, completely denied. During the last three decades, access to genetic diversity and benefit-sharing conditions has become more complicated and bureaucratic. Given the fact such access is an essential requirement for any breeder, breeders started to pay more interest in strengthening their own working collections. Eventually, seed companies established their own private genebanks to ensure their in-house breeders have permanent access to the genetic resources they need for developing new varieties.

In parallel to the above developments, many public genebanks, especially in developing countries saw increased budgetary constraints, faced an increasing lack of required expertise and equipment, and thus became increasingly isolated from the global user community [1]. The mentioned legal complexity of providing access to healthy and good quality genetic resources further contributed to a weakening position of part of the public genebanks in offering targeted services to society and consequently resulting in less support for investments in conservation and use.

The above situation has prompted the authors to review the current practices of collecting, conserving, and maintaining germplasm accessions more systematically and critically as well as how these genetic resources are accessed, in particular by the private breeders and their breeding companies, as their role in conservation is steadily increasing, albeit restricted largely for their own use. Therefore, identifying the bottlenecks and constraints in the current global conservation and use system that impact the collaboration between private and public genebanks is the first step towards resolving them. In this review paper, we will assess germplasm flows, procedures, traditions, and other (legal) processes that might include such constraints, bottlenecks, and possibly other negative reasons with respect to the collaboration, followed by presenting possible solutions. This paper intends to contribute currently missing information to the political debate on revising global legal instruments and national/local conservation practices in order to strengthen the effectiveness and efficiency of germplasm conservation and use efforts. 

### 1.2. Why Focus on Vegetables?

Breeding crops is a multidisciplinary activity that strongly varies in methodology from one crop to another. The way that genetic resources are being used in the breeding process, therefore, also varies significantly between crops or groups of crops. The latter depends, among others, on the ‘development state’ of the crop or group of crops, including to which extent they have been researched. To avoid unnecessary complexity in our assessment of the situation of access, it was deemed easier to restrict the overall range of crops. A somewhat arbitrary decision was made to focus this paper on vegetable crops, whenever possible, meaningful, and applicable. Only a relatively small number of known public genebanks had a strong collaboration history of working with the private breeding sector. Also, only a few curators from private genebanks and only a few private plant breeders had expressed preparedness to share relevant information for this paper. Consequently, this paper has a rather strong focus on a limited number of countries and genebanks (see also Section 2). When no specific information on vegetable genetic resources was available, ‘general information’ was used with the assumption that this would also apply to vegetable crops. 

Accessing new genetic diversity is essential in the breeding process of any vegetable crop and there is, in general, a broad range of genetic diversity available from public and private genebanks for these breeding activities. However, many ’minor’ vegetable crops are underrepresented in genebanks, which is typical for the neglected and underutilized species—NUS. Among the major vegetable crops, only the Brassica complex (Brassica et al.), carrots (Daucus), and eggplant (Solanum) form part of Annex I crops of the International Treaty on PGRFA, while all the minor ones are not included. Furthermore, in particular, the minor vegetables are, in general as well as in breeding terms, less ‘advanced’ compared to the major staple crops as many of the minor vegetable crops have hardly undergone any significant breeding. This means that we have a vast spectrum of vegetable crops with respect to their improvement status and thus, the vegetables at large represent the entire spectrum of their dependency on genetic diversity for the breeding process in creating new, more nutritious, and better-adapted varieties.

Whereas vegetable genetic resources are only a smaller proportion of the total range of PGRFA, the findings on the collaboration between public and private genebanks of vegetable genetic resources are representative of all PGRFA and all genebanks, and very suitable for illustrating the points that are being made. This same line of thought also applies to the limited number of genebanks and countries that provided information, the findings can be extrapolated to other genebanks as well, without too many restrictions.

Some general considerations

While attempting to strengthen the collaboration between public genebanks and breeding companies, one must consider that such collaboration should benefit both sides. As already mentioned, the ‘business model’ of breeding companies is to breed new varieties that address problems experienced by growers and meet the expectations of consumers. Newly bred varieties need to be competitive and possess environment-friendly properties, thus being sustainable and ‘marketable’, generating financial benefits to the companies, besides other benefits to the society at large. To do so, the companies need access to plant genetic resources.

Public genebanks aim at effective and efficient long-term conservation of defined national and/or economically important crop genepools and to make these genetic resources available to users. They often also have the mandate to contribute to the conservation of the bio-cultural heritage of their country and region. Genebanks tries to make the conserved accessions available in the most user-friendly form, along with related information. Furthermore, they should assist in the acquisition and research of genetic resources of importance to their national agricultural economy. Many of the (national) public genebanks are well-positioned to facilitate access and benefit-sharing arrangements. Fulfilling all these requirements will allow them to significantly contribute to national and global food and nutritional security and a more sustainable form of agriculture. 

## 2. Materials and Methods

Considering the (political) ‘sensitivity’ of this subject and the hesitation from the breeding companies to openly engage in assessing and critically reviewing the current practices concerning the conservation of genetic resources in their companies, it was decided to base this review on the information that had been received from a limited number of private-sector breeders and company genebanks, including Rijk Zwaan, the Netherlands, and East-West Seed International Limited, Thailand as well as from the literature. The inclusion of data from the Dutch national genebank CGN is important as it has a significant collection of different vegetable crops is one of the most advanced national genebanks in the world with respect to genebank and germplasm management practices, and has gathered significant experiences in collaborating with breeding companies in the Netherlands and beyond. Another national genebank with a very broad portfolio of field and vegetable crops and their related wild relatives is the Leibniz Institute of Plant Genetics and Crop Plant Research (IPK), Gatersleben, Germany. It is among the most important national genebanks that distribute germplasm worldwide and is, similar to CGN, one of the most advanced genebanks. The only global vegetable genebank is the World Vegetable Center (WorldVeg), with headquarters located in Taiwan. It is the 5th biggest international public genebank with a unique focus on vegetables. It is closely associated with the CGIAR genebank platform and has established significant collaboration with private and public vegetable breeders and companies. The long and extensive experience of the authors in operating public genebanks, including the conservation and facilitation of the use of a wide diversity of (vegetable) crops and their wild relatives, their experience in collaborating with the private genebank/breeding sector as well as with the framing of the global, regional and some national legal frameworks and applying it on a routine basis, provide a good foundation for this review and to identify and present a number of opportunities that would facilitate and strengthen the collaboration between two complementary activities, i.e., conservation and use.

## 3. A Historical Overview of Accessing and Managing Germplasm by the Private Sector

Since the first steps of domesticating plant species were made some 10,000 years ago in the Fertile Crescent, and even earlier when gathering wild plants, human selection started to play an essential role in shaping the foundation for agriculture. Farmers began to select species and genotypes within species for cultivating ‘crops’. This process lasted for millennia and continues even today. It was not until after the rediscovery of Gregor Mendel’s principles of inheritance, published in 1865 [2], that scientific breeding based on crossing and selection started at the beginning of the 20th century. Plant breeding as a commercial undertaking is much older, possibly dating back to the start of breeding efforts by de Vilmorin in France in 1743 [3]. Ample genetic diversity is the foundation for selecting a suitable population or the best genotypes within a given species for cultivation, or parents for making crosses and thus to produce varieties that stack desirable characteristics. During the first part of the 20th century, plant introduction centres were established for single crops, like potatoes, or multiple crops [4]. To ensure the continued availability of sufficient genetic diversity to select from, breeders typically establish working collections of genotypes of a given crop that would correspond with their breeding objectives [5].

At a later stage, targeted collecting of genetic diversity of a given crop, in the form of landraces, traditional farmers’ varieties, or related wild relatives of such crops, was initiated by plant breeders and scientists, and the first genebanks were established, mainly for research purposes, first in the USA (Harry Harlan) and Russia (Nikolai Vavilov), later in Germany (Hans Stubbe) and in other European countries [4]. However, with the occurrence and increase of the so-called genetic erosion, especially following the large-scale introduction of scientifically bred varieties with a limited genetic base [6], the foundation of crop improvement and plant breeding started to become threatened, and thus, the earlier priority of public genebanks to provide genetic diversity to users changed to a more conservation orientated objective. Consequently, efforts were undertaken to more systematically collect and conserve genetic resources, at the beginning especially landraces of crops, by and in genebanks worldwide. 

Until the early 1990s, access to the genetic resources available in situ or conserved in the predominantly public genebanks was free and unrestricted. This access paradigm was embedded in the International Undertaking that FAO had concluded as a voluntary international agreement with their member countries in 1983 [7]. Thus, breeders had ready access to genetic diversity conserved in genebanks and/or available in farmers’ fields and natural populations. Generally, this on-farm and in situ germplasm could be collected without major restrictions. As these materials were typically added to or shared with the public genebank collections in the respective countries, and as genebanks exchanged germplasm among them to respond to requests of users/breeders, only limited access problems existed. Mostly public breeders organized collecting missions, frequently together with scientists from universities and/or genebank staff and added collected material to their working collections. Through collaboration with research institutes and universities, private breeders had easy access to these collections and pre-bred materials. For instance, in the Netherlands, public research institutes had a strong focus on the production of pre-bred materials that could be made readily available to private and public breeders [8]. From all these sources, including from public genebanks and research institutes worldwide, germplasm materials and advanced breeding lines have been incorporated into the working collections of private breeders. 

Private breeders, in general, prefer to use commercially successful ‘elite’ varieties of a given crop as parents in their breeding activity and ‘only’ resort to genetic resources, in particular, landraces and crop wild relatives, when they need to enlarge the genetic diversity pool and/or want to include specific traits in the ‘elite variety’ they plan to produce [9]. In particular, the use of crop wild relatives in breeding programmes generally requires lengthy and costly back-cross programmes to obtain genotypes/varieties that no longer possess unwanted characteristics from the wild species. In this context, it should be noted that genomic tools such as marker-assisted selection allow a much more effective removal of linkage drag. Furthermore, genome-wide association mapping and comparative genomics allow the identification of marker-trait associations and the prediction of associated candidate genes. This information can then be used for the efficient incorporation of desired traits through marker-assisted breeding (see, for example, Puranik et al. [10]). It should be stated that the above arguments apply to vegetable genetic resources and PGRFA at large.

To give breeders protection for their efforts, a system of intellectual property protection of released varieties, so-called plant breeders’ rights, was set up in the second half of the 20th century. The Union for the Protection of Plant Varieties or UPOV is the most widely applied ‘system’, which gives breeders for up to 20 years the monopoly of reproducing and selling the protected variety. Such protected varieties can still be freely used by competing breeders for further breeding activities under the so-called ‘breeders’ exemption’. In this context, also the US 1930 Plant Patent Act should be mentioned as it aimed to protect newly released varieties through a patent. Through the ‘harmonization’ process of the International Undertaking and the CBD the principle of ‘common heritage of humankind’ was eliminated and replaced by the notion of ‘national sovereignty’ of states over their natural resources [7]. This process of increased application of legal protection of varieties, especially after the application of patents on genes, traits, and even varieties, resulted in more restrictive accessibility of genetic resources, especially from ‘the Global South’ and was further fueled by the establishment of the Convention on Biological Diversity (CBD) in 1992, including the recognition of ‘national sovereignty’ of states over the biological resources in their territories and in 2014 by its related Nagoya Protocol. Subsequently, the FAO had to counter the access and benefit-sharing challenges those bilateral negotiations under the CBD posed to the exchange of plant genetic resources for food and agriculture. This process resulted 2001 in the establishment of the International Treaty on Plant Genetic Resources for Food and Agriculture (hereinafter called ITPGRFA or International Treaty), which entered into force in 2004 [11]. In harmony with the CBD, it made specific arrangements for PGRFA, notably a multilateral system for access and benefit-sharing [4].

The CGIAR genebanks contribute(d) significantly to the collecting, acquisition, and long-term conservation and use of PGRFA, maintained in the public domain for the entire global community, especially regarding staple crops. The developments over the past 30 years with respect to the acquisition and distribution of germplasm by the CGIAR centres were analysed to demonstrate the current situation with respect to ABS by Halewood et al. [12]. A highly political environment was observed, such as countries’ unwillingness to share their materials. Restrictive national laws and policies were the reasons most often cited by the CGIAR centres’ genebank managers for decreased rates of acquisition of additional materials to conserve in and distribute from their genebanks. Furthermore, the following more specific scenarios that impacted negatively on the acquisition of germplasm by the CGIAR centres were reported: a combination of different elements, including ABS aspects, that have built mistrust on geopolitical levels; intentions or promises to share materials by (technical-level) partners that have been thwarted by political arguments; insecurity on the part of national partners, in particular in the Global South, because of unclear lines of authority and fear of being accused of ‘selling out’ the country’s patrimony; insecurities on the part of the centres to even request for materials, given the vagaries of national procedures and the possibilities of backlashes and a considerable degree of uncertainty throughout the entire national and international system [12].

In a more recent study regarding rates of acquisition and distribution of germplasm materials by the eleven CGIAR genebanks, Halewood et al. [13] observed an increasing geopolitical polarization over access and benefit-sharing arrangements, as well as the unwillingness of several International Treaty member states to share germplasm through the multilateral system as they feel that they do not get sufficient recognition for their germplasm maintained and shared by/with the centres. Furthermore, developing country contracting parties are dissatisfied with the fact that only three payments have been made to the Plant Treaty’s Benefit-Sharing Fund by commercial users of materials from the multilateral system, two back in 2016 [14] and one in 2018 [13]. It may be assumed that all this is true for vegetable collections as well. Although benefit-sharing arrangements are not the main focus of this paper, examples of actual or perceived benefit-sharing arrangements are included in Section 5.5.

As a result of the decreasing collecting and access to PGRFA, breeders support joint collecting missions with public genebanks, implemented within the provisions of the existing legal framework. Collecting is predominantly done in centres of diversity and has a focus on landraces and crops of wild relatives. Breeders also increasingly established their own working collections, particularly through the acquisition of germplasm accessions from national and international genebanks, as well as pre-bred materials from research institutes and commercial varieties worldwide. Gradually, these breeding collections increased in size. With the introduction of more complex and restricted access regulations by countries and genebanks, many companies decided to start their own genebanks to maintain this in-house germplasm for the long term, assuring continued access to the basic material of their breeding activities. This will be further elaborated in Section 5.

The traditional way of (private) breeders to acquire the needed germplasm materials either from or with the assistance of a national public genebank or by joining collecting missions to centres of diversity, has become more complicated and bureaucratic. This also applies to the sharing of pre-bred materials among national plant research institutes and private breeders. At the same time, and as a response to the general trend, we observe increasing numbers of consortia, including partners from public genebanks and private breeding companies that facilitate the sharing of selected accessions among the participating institutes.

## 4. Why Do Breeding Companies Establish Genebanks?

Considering the developments concerning access to plant genetic resources, an increasing number of breeding companies decided to expand their conservation efforts by consciously adding genetic diversity to the existing breeding collections of crops that are part of their breeding activities and thus, establishing their own private genebank for long-term conservation. The first mention of breeding companies to have established their own genebanks for some crop species was made by Kate and Laird [15]. However, this development process of establishing genebanks by companies started much earlier but was possibly not published or reported [9]. In this section, we elaborate on the reasons why this happened. The content presented is based mainly on the information provided, through personal communication, with a managing director and a genebank manager, i.e., Kees Reinink of Rijk Zwaan [3] and Marilyn Belarmino of East-West Seed International Limited [16], respectively as well as on the knowledge and experience of the three authors. 

The main reasons to establish a genebank by breeding companies have been grouped into two parts: (1). Cost and efficiency considerations; and (2). Securing future access to needed genetic diversity.

### 4.1. Cost and Efficiency Considerations

Breeding companies were already managing large working/breeding collections, sometimes of several hundred thousand samples, in the case of larger seed companies. Therefore, adding ‘a few ten thousand’ accessions acquired by purpose from different sources to increase the diversity would neither create a significant additional cost nor significantly increase the workload. Furthermore, having your own genebank with a good representation and documentation of the genepool(s) concerned contributes to considerable time and cost savings by not being forced to acquire or collect the required genetic diversity each time when needed.

When starting a screening process, it is advantageous to have the entire range of genetic diversity of a given crop already ‘in-house’. Especially having sufficient seed quantity of the accessions to be screened saves considerable time and avoids the need to request new/additional germplasm materials, usually requiring extensive correspondence and time. When participating in research or breeding projects, or regional/global consortia, the sharing of germplasm is typically a pre-condition, and this requirement can be more easily met when having an own genebank.

Many national and institutional genebanks in less developed countries are operating on limited budgets and this frequently results in the distribution of poor-quality germplasm samples that often are not yet sufficiently characterized. Furthermore, the amount of seeds/plant propagules per sample provided by public genebanks is always limited, requiring a seed or tissue multiplication step before screening can be initiated, adding to cost and loss of time. Another, albeit less frequent comment related to the quality of the distributed germplasm is that the genetic composition of individual accessions might not meet the expectations of the recipients. For many of the evaluation activities, breeders need uniform germplasm samples. Especially when the requested material is used for molecular activities, the availability of accessions consisting of single seed descents would greatly facilitate their ‘instant use’. However, many of the traditional genebank accessions are, heterogeneous, and therefore require an additional step to obtain uniform samples for screening or evaluation activities. Related to this, public genebanks are frequently unable to respond, in a targeted manner, to trait-specific requests made by breeders for germplasm accessions and/or might not have the supporting characterization/evaluation data of the provided accessions at hand.

The phytosanitary status of germplasm accessions is an issue of increasing concern. Materials from public genebanks often contain seed-transmitted infectious pathogens. This might be caused by the fact that many genebanks are not able to keep their collections disease-free, because of high costs or a lack of adequate and up-to-date seed health testing facilities and expertise [17]. Once the germplasm material has been received by a private company genebank, the accessions/samples must be tested and cleaned, often a requirement of the national phytosanitary authorities, and thereafter maintained clean.

### 4.2. Securing Present and Future Access to Needed Genetic Diversity 

Possibly the most critical reason for having an own company genebank is to ensure that good quality and securely conserved genetic resources are readily available to the breeders to support pre-breeding, breeding, and research programmes, now and in the future [18]. Over the past years, it has become increasingly more challenging to acquire germplasm from national or local genebanks. The most important reason for this is that most genebanks have a policy to only provide a limited number of accessions per request and/or year and requester. Another reason is that several public national and many institutional genebanks that frequently manage crop-specific collections, lack a functional information management system and thus are not able to effectively deal with germplasm requests. A very different reason is that used material transfer agreements sometimes include a requirement that the obtained germplasm should be destroyed after a single use or a demand for high royalties upon the release of a commercial variety. 

The current complexity of policy and legal instruments regulating ABS arrangements for germplasm is possibly the most critical reason for breeding companies to establish and operate their own genebanks. In addition to current bureaucratic access restrictions, there is also a great level of uncertainty around the ongoing political debate concerning future ABS regulations triggered by the diversity in national legislation to address the requirements of the Nagoya Protocol [5]. This high degree of uncertainty about future germplasm access has motivated breeding companies to establish their own genebanks and to acquire a wide range of potentially useful genetic resources of company-specific target crops from public genebanks or through collecting missions. It should be noted that since the late 1980s discussions ensued about the application of property rights over newly bred varieties, typically protected through plant breeders’ rights and increasingly with patents on plant traits and the underlying genes, but also varieties). As many developing countries and NGOs have opposed these developments, this issue has certainly contributed to critical views in the private sector and consequently, becoming more restrictive in providing easy access to genetic resources. 

Some more specific points related to international, regional, and national ABS legislations and regulations, that have been raised in this respect include: Seed companies decided to maintain all accessions that had been obtained in the past from third parties in-house for long-term conservation to prevent having to do all ‘the burdensome paperwork again and again’. Over the past years, it has become more difficult to import germplasm due to burdensome pre-shipping requirements, partly also due to the increased risk of transboundary spread of pathogens and insect pests [19];Typically, companies carefully study the conditions included in contracts and SMTAs from public genebanks and other sources that must be signed before accepting genetic material. If the conditions are unacceptable, especially when demanding unrealistic benefit-sharing requirements, such material will not be included in the company’s breeding pool. In this context, the Nagoya Protocol should be mentioned as an important ‘trigger’ for these developments;An important legal aspect of the current benefit-sharing arrangements for a breeding company is that they oppose everlasting obligations. These are often included in ABS rules and require that the company that does the introgression of an interesting trait from a genebank accession into breeding material has to pay as long as this trait is present in one of its varieties, whereas all competitors, who just take the trait from a released variety, based on the breeders’ exemption, do not have to pay ABS. This rule thus puts the company that does the largest breeding effort in a disadvantageous position.

The expectation that breeding companies share evaluation information on the acquired accessions with the providing genebanks is for many companies difficult to accept. They prefer not to share their internal evaluation results or, when participating in research or other consortia, agreements often include embargo periods before screening results are publicly shared. Thus, having your own germplasm accessions from the ‘private’ genebank circumvents such requirements.

In summary, the reasons for breeding companies to establish their own genebank are manifold. In many instances, they are directly related to the fear of having to spend increasingly more time and money to obtain sufficient and good quality accessions along with the relevant information from genebanks around the world, or, in some cases, no longer being able to access those resources at all.

## 5. Practices of Breeding Companies to Acquire Germplasm 

As already stated before, breeders have their own working collections for each crop they work on. These collections are typically dynamic, and they reflect the actual breeding objectives and priorities for a given crop. Many collections also include traits in response to current and future demands from their direct customers, farmers, and the market for new varieties. Also, useful breeding materials from the own programme, released varieties from competitors, as well as research materials, can be found in these working collections. 

For a correct interpretation of the current global situation with respect to access and benefit sharing, it is important to review how private breeders have traditionally obtained the genetic resources for their breeding programme and, more recently, for the conservation and use of their own genebanks. The most common way has been through formal acquisition from public national or international (e.g., the CGIAR and WorldVeg) genebanks, as will be presented below. Another important way was/is through participating in collecting missions within the country where they are based and/or abroad. The latter is usually through collaborative projects with the (national public) genebank and/or with public or university research teams. It is typically also through these arrangements that possible benefits are shared. Furthermore, the acquisition of pre-bred materials has been and still is a relevant source of diversity to the private sector breeding and genebank activities. Today, public-private partnership (PPP) research consortia and networks are also sources of germplasm materials shared among and by partners. Unfortunately, it is impossible to quantify these different ways of germplasm acquisitions by private genebanks. Details on the acquisition sources and the common practices of the private sector to obtain the genetic resources they need are elaborated further below. 

Regarding the acquisition of germplasm from genebanks, especially national genebanks, a few aspects should be mentioned as background information. Genebanks have included the distribution of germplasm materials they conserve to users as one of their routine operations and responsibilities. In general, public genebanks are well connected to the various research and breeding institutions of the agricultural sector in their country and abroad. Recently, Mekonnen and Spielman [20] correlated historical trends in genebank acquisitions and changes in germplasm exchange over time, with changes in national and global policy environments for seven crops (sorghum, cowpea, pearl millet, beans, maize, rice, and wheat) that are essential for food security in developing countries. Based on these results, the authors concluded that a sharp decline in genebank acquisitions was observed in 1993 when the CBD entered into force and that country’s membership in the CBD is closely associated with reductions in the flow of genetic resources. Furthermore, the Nagoya Protocol may affect global PGRFA flows in a potentially negative and unintended manner. In contrast, ITPGRFA membership is likely to moderate the adverse effects of the CBD and the Nagoya Protocol [5]. 

Since the implementation of ABS regulations, the distribution of requested materials is done under material transfer agreements. In the case of species listed in Annex I of the International Treaty, often the standard material transfer agreement (SMTA) is used [21], and an increasing number of countries are using the SMTA also for germplasm materials not listed in Annex I, e.g., the Netherlands, Germany, and most Nordic countries [5]. The following paragraphs will provide examples of the distribution of predominantly vegetable genetic resources by the Netherlands, Germany as well as the international genebank of WorldVeg.

### 5.1. Germplasm Distribution by Selected National Genebanks

In the ten-year period 2013–2022, CGN distributed 46,819 samples or an average of 4620 samples per year, of which 22,996 (=49%) samples had been sent to private, predominantly breeding companies. The latter included 11,634 (=25%) samples that were sent to companies outside the Netherlands. Most of this material concerns vegetable seeds of landraces and crop wild relative species (Figure 1).

**Figure 1 plants-13-00247-f001:**
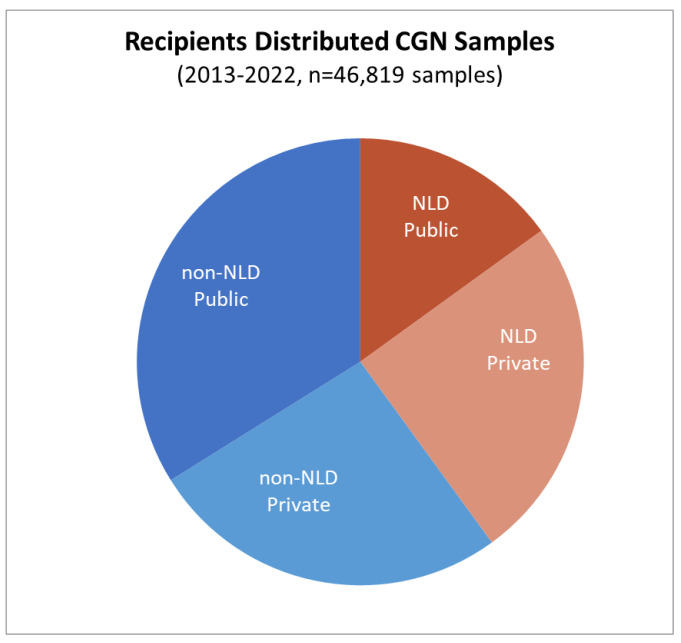
Germplasm distribution from the Dutch national genebank CGN from 2013–2022 to recipients worldwide. ‘NLD’ stands for the Netherlands, ‘Public’ refers to public institutions including universities, research organizations and genebanks, and ‘Private’ to (mainly) breeding companies. To illustrate the development with respect to the distribution of genebank accessions/samples to users for all crops and species, distribution data have been obtained from CGN since its establishment in 1986. The number of annually distributed samples and answered requests are presented in Figure 2. CGN makes its material available to any user who wants to use it for research, breeding, or education, and distributes its material all over the world. All distributions are done under the SMTA. Sometimes the phytosanitary or other import requirements make shipment difficult or impossible.

**Figure 2 plants-13-00247-f002:**
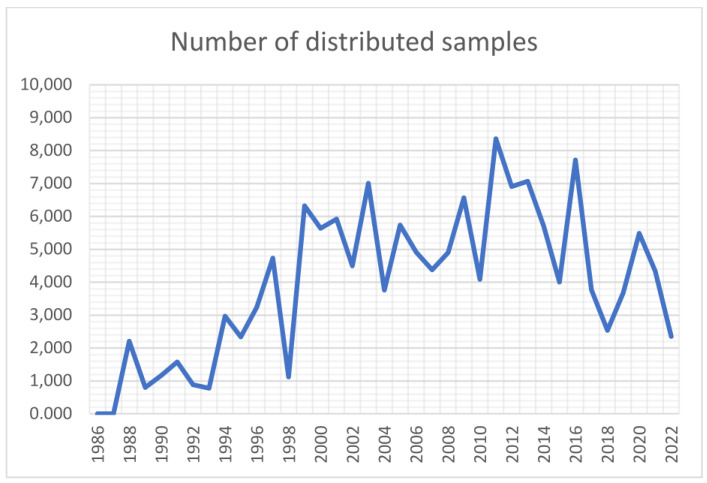
Number of annually distributed germplasm samples by CGN since its establishment in 1986.

CGN has the policy that all requests for more than 50 accessions need to be questioned by its curator of the corresponding crop genepool; usually, this process results either in a reduction of the number of accessions being shared in cases where a better selection of accessions or traits can be made, or in an agreement about sharing the results in cases that large scale screenings are done. The screening results are only made public after a negotiable embargo period of about three years. The fluctuations in the number of distributed samples over the years can largely be explained by incidental large-scale screening projects, resulting in an inflation of the number of distributed samples. 

As illustrated in Figure 3 and based on data received from Weise and Oppermann [22], IPK distributed during the period 2006–2022 a total of 142,157 vegetable germplasm samples which is 34.0% of the total of 418,197 samples distributed of all crops and species during the same period. It should be noted that the data on vegetables are based on an internal IPK crop classification (55 crops or crop groups, of which 20 were regarded as vegetables) that required in some instances arbitrary decisions to include or not include a given category as a vegetable. Two examples are the inclusion in this study of *Phaseolus vulgaris* and *Solanum tuberosum* as vegetables. Whereas 14.5% of the total distributed 418,197 germplasm samples were sent to German (4.4%) and foreign plant breeders (10.1%), of the 142,157 samples of vegetable germplasm 5.6% were sent to German plant breeders and 11.4% to breeders outside Germany. The 87,214 samples of vegetable germplasm distributed to all German recipients (i.e., 61.4% of the total of 142,157 samples) was more than 1.5 times higher compared with the total samples distributed to all recipients outside Germany, i.e., 54,943 samples or 38.6%). 

The distribution figures of the Dutch and German national genebanks of all germplasm accessions maintained are impressive and seem to have reached a maximum in 2011 in the Netherlands (almost 8500 samples) and in 2015 in Germany (almost 45,000 samples). The distribution rate to the private sector, predominantly plant breeders, is roughly the same for Dutch and foreign breeders. In Germany, the distribution of all germplasm to foreign breeders is slightly lower (43.9%) than the distribution to German breeders (i.e., 56.1%). The percentage of samples distributed to breeders from all the germplasm distributed during the period 2006–2022 by IPK is 14.5%, with a variation from 9.4% in 2015 to 23.9% in 2019.

Day-Rubenstein et al. [23] reported that about 5% of the 162,673 germplasm samples distributed by the United States National Plant Germplasm System from 1995 to 1999 went to commercial companies outside the United States. 

It is interesting to note that the distribution data stems largely from genebanks and countries that have very limited restrictions on the distribution of public germplasm. Unfortunately, several other ‘big genebanks’, e.g., the Indian, Chinese, and Russian national genebanks, have a far more restrictive policy regarding the international distribution of publicly conserved germplasm.

### 5.2. Germplasm Distribution by CGIAR and WorldVeg Genebanks

Regarding germplasm acquisition from genebanks by the private sector, one of the very few information sources is the State of the World reports that FAO produces and publishes about every decennium. These reports are based on country reports produced by the formally designated focal institution (usually the national genebank) that gathers the information nationwide. For this reason, the information in the reports is predominantly related to the public genebanks [1,24,25]. The majority of germplasm accessions used in public and private plant breeding were sourced from national genebanks (more than half), followed by CGIAR genebanks and international and regional networks (about one-third), local genebanks, public institutions from developed and developing countries, and the private sector [1]. During the period 2017–2019, the following CGIAR centres were reported to have distributed germplasm to the commercial (mostly breeding) sector: Alliance Bioversity and CIAT, CIMMYT, CIP, ICARDA, ICRISAT, IITA, and IRRI. Seven percent of the approximately 200,000 samples were sent to the commercial sector [13]. 

Regarding the type of germplasm, some data are available for the period from 1996 to 2006, during which the international agricultural research centres distributed a significant amount of germplasm to the private sector; 51.7% of the distributed accessions were landraces, 36.0% breeding lines, 7.1% crop wild relatives and 5.1% improved varieties [25]. The latter coincides roughly with the data reported by Halewood et al. [13] for the period 2017–2019, i.e., 50% landraces, 24% breeding materials, 13% crop wild relatives, and 6% improved varieties.

More detailed but somewhat dated information from WorldVeg illustrates a more detailed picture of the distribution of vegetable genetic resources. This international nonprofit institute for vegetable research and development actively exchanges genetic resources and related information with national programs, regional organizations, and the private sector. As recent distribution data are not accessible, we refer here to data published earlier, giving a reliable overview of germplasm distribution until about 2014. Until this period, the WorldVeg genebank distributed approximately 6000–7000 accessions and breeding lines each year for crop improvement programs and related research worldwide. Although no detailed figures are available, the ratio of ‘pure genebank accessions’ to ‘improved breeding lines’ is approximately 20 to 80% for the vegetable crops for which WorldVeg operates crop-specific breeding programmes. In contrast, this ratio is roughly the reverse for minor vegetables and indigenous/traditional crops without a WorldVeg breeding program. Based on 2012 data, the largest number of annually distributed seed samples is shared in-house with breeders and other scientists, e.g., virologists, entomologists, and molecular biologists (37%), followed by national agricultural research and extension systems (26%), breeding companies (22%), universities (10%), non-governmental organizations (3%), and others (2%) around the world [26]. Based on 2012 data, the efforts of WorldVeg breeders, focusing on several major vegetable crops as well as on traditional crops of more local/regional importance, such as amaranth, African eggplant, okra, roselle, Corchorus, and bitter gourd [27], led to the accumulated release of more than 466 improved vegetable varieties, developed with/using WorldVeg germplasm and breeding lines, around the world [28]. 

Chilli pepper (*Capsicum*) is the most widely distributed crop by the WorldVeg genebank, followed by tomato. During the period from 2001 to 2012, 29,980 germplasm materials were distributed, comprising 6008 genebank accessions (20%) and 23,972 advanced breeding lines (80%), developed by WorldVeg breeders [29]. The top ten recipient countries of Capsicum germplasm during that 12-year period were India (4671; 15.6%), the Republic of Korea (2710; 9.0%), Thailand (2484, 8.3%), China (2416, 8.1%), USA (2211, 7.4%), Vietnam (1418; 4.7%), Taiwan (1154; 3.8%), Indonesia (1064; 3.5%), the Netherlands (717; 2.4%), and Tanzania (509; 1.7%). The National Agricultural Research and Extension Systems (NARES) received the largest germplasm share (13,672 samples; 57.0%), followed by breeding companies (9741; 32,5%), universities (4558; 15.2%), private individuals (1611; 5.4%), and non-governmental organizations (NGOs) (398; 1.3%).

An analysis of tomato distribution data, the second most widely distributed vegetable crop by the WorldVeg genebank after *Capsicum*, was undertaken during the period from 2001 to 2013 [30]. During that 13-year year period, a total of 27,438 germplasm samples and breeding materials were distributed from headquarters to 138 countries worldwide. Resembling the Capsicum distribution data, the majority of the distributed seed samples (22,258; 81%) were improved lines developed by WorldVeg breeders, while 5180 seed samples (19%) were genebank accessions. 

It is quite interesting to note that the top three tomato-producing countries in the world (China, India, and the US) were among the top four tomato germplasm recipient countries of the WorldVeg genebank—an indication of the relevance of WorldVeg tomato germplasm/advanced breeding materials for the top global producer countries. Most countries preferred to receive advanced lines that had been developed by WorldVeg breeders, except for Japan (due to a high demand by local breeding companies to address particular local demands) and Pakistan. For these two countries, the share of genebank accessions clearly dominated and reached 81.8% and 60.6%, respectively [30]. The Netherlands also had a relatively high share of genebank accessions of 41.5%. This is an indication that those countries have strong public and private tomato breeding programmes, capable of exploiting the full potential of the genetic variability of germplasm accessions. The share received by the different categories of users is similar to *Capsicum* distribution data. Government organizations (10,601 seed samples; 39%), breeding companies (8882 samples; 32%), and universities (5873 samples; 21%) were the top three recipient categories of WorldVeg tomato germplasm.

Even though the above distribution figures are somewhat dated, these data are still a good indication of the distribution of WorldVeg advanced breeding materials and genebank accessions to the private sector. It can be assumed that the distribution pattern remained similar until 2019/2020, with COVID-19 most likely causing significant disruption in the flow of germplasm and seed supply chains, especially in the Asia-Pacific region but also in Africa [31,32]. Whereas no data on the distribution of minor vegetable crops by WorldVeg are available, it should be noted that significant germplasm distribution of minor crops, in general, is related to breeding efforts by WorldVeg breeders, either based in Taiwan or Africa. A quote from WorldVeg’s [33] strategic breeding plan describes the crop-specific breeding approach to achieve impact: “*It is important to recognize that impacts were achieved through very different pathways and partnerships depending on crop and location. There are contrasting impact pathways between (open-pollinated) varieties and hybrids as well as between countries with developed and underdeveloped seed systems. For instance, WorldVeg breeding lines of tomato and chilli pepper made a large impact through private sector pathways but made very little impact through public sector pathways. In contrast, mungbean breeding lines created tremendous impact through public sector pathways and negligible impact through private sector pathways. This shows that the Center needs to be strategic in how to tailor its breeding products to achieve impact at scale*”. The above distribution figures from WorldVeg beg the question about benefit-sharing. One important dimension of sharing benefits with the ‘countries of origin’ of the collected germplasm is the facilitation of the distribution of newly bred varieties to farmers and local plant breeders, particularly in developing countries that provide the original germplasm sources. Another dimension is the sharing of information from the private plant breeders back to the genebank and breeders at WorldVeg and thus, at least indirectly, contributing to the sharing of benefits. However, it should be noted that feedback from private breeders is difficult to obtain, whereas public plant breeders tend to be more willing to provide feedback information on the performance of shared germplasm and breeding lines.

The analyses of germplasm distribution data from the national genebanks of the Netherlands and Germany show a significant and steady flow of germplasm from the public genebanks to breeders and breeding companies. Whereas in the case of CGN, 49% of the distributed samples between 2013 and 2022 (annual average 4620 per year) went equally to breeding companies in the Netherlands and abroad, in Germany, 14.5% and an average of 3573 samples per year of all crops/species distributed went to breeders, both public and private, of which about 1.5 times more to breeders abroad. For vegetable germplasm, these figures are 17.0% to plant breeders with an average of 1418 samples per year. The distribution by the CGIAR genebanks fluctuated from year to year and was slightly over 4200 distributed samples in 2019 to the private sector. WorldVeg distributed between 6000 and 7000 samples of germplasm accessions and breeding materials annually. In 2012, 22% of the distributed samples went to breeding companies. It should be noted that in all analysed cases, the distribution reached a maximum between 2011 and 2016.

### 5.3. Germplasm Collecting

Regarding collecting missions implemented or supported by breeding companies during the last twenty years or so, only very limited information is available. In general, collecting missions have not been organized and undertaken by breeding companies, in some cases the latter supported such efforts financially. Most information on joint collecting efforts has been obtained from the Netherlands. Collecting missions have been undertaken by CGN, with the financial support of the breeding companies, and by some breeding companies themselves, especially in countries situated within the centres of diversity of a given crop, since the early 1900s. The focus was mainly on local landraces, only during the second half of the last century crop wild relatives were gradually included.

Since its establishment, CGN has been actively collecting PGR material. An overview of all collecting missions it organised is given at https://missions.cgn.wur.nl/; accessed on 23 August 2023 [34]. Details on collecting missions implemented over the past ten years with support from private vegetable breeding companies are presented in Box 1.

Box 1Collecting missions organized by CGN with the support from vegetable breeding companies over the past ten years.
The Centre for Genetic Resources, The Netherlands has been actively collecting PGR material since its establishment. An overview of all collecting missions it organised is given at [33].The most recent missions, for which detailed information is available, include:2013 Armenia and Azerbaijan115 wild populations of 7 *Lactuca* species2015 Uzbekistan and Kyrgyzstan190 *Daucus* wild populations and 22 carrot landraces2017 Uzbekistan50 melon landraces2017 Jordan51 *Lactuca aculeata* wild populations, 1 *Lactuca*     *serriola*, 1 *Lactuca saligna*, 1 *Lactuca undulata* and 1 *Lactuca orientalis* population2019 Uzbekistan21 *Lactuca altaica* wild populations, 28 *L. serriola* populations and 13 mixed populations.


The above joint collecting efforts between a public genebank and private companies stem from one country and are certainly not representative of the global picture. However, it can be noted that the Dutch National Genebank is a good example of how collaboration with breeding companies can contribute to successful conservation efforts, both in the countries where the collected (through training and the deposit of half of the collected samples in the national genebanks) as well as in The Netherlands. However, as can be observed from the above box, almost all collecting missions are conducted in Central Asia and on a limited number of crops. Due to the impact of the CBD and the Nagoya Protocol, no missions have been possible to collect for instance germplasm of the major vegetable crops in Latin America and some countries in Asia.

### 5.4. Germplasm Exchange through PPPs and Other Research Consortia

As mentioned before, research consortia as public-private partnerships have been established since approximately 2000, frequently under the coordination and execution of a public institute together with variable private sector entities. It should, however, be noted that also during earlier periods, public-private partnerships were established, for instance, to ‘hunt’ for rare plant materials during the 18th and 19th centuries. In activities that focused on plant breeding, germplasm materials were either resources from public genebanks only or partners were expected to share some of their germplasm. As part of a European Cooperative Programme for Plant Genetic Resources (ECPGR) initiated project on PPP activities, a small database on projects has been established, including details on germplasm source acquisition [35]. PPPs are regarded as a possible approach to address market failure in the field of technology innovation when the public and the private sectors are not able to carry out the required R&D activities on their own. In recent years, many PPPs in plant breeding have been established. Among those is a PPP initiative of the Nordic Council of Ministers for pre-breeding activities in the Nordic countries. The PPP initiative was proposed in 2010, and the first call for proposals was launched in 2012 [36]. The PPP is based on pooled public funding, project-based participation of interested plant breeding companies, engagement of state-of-the-art research facilities for the respective projects, and an equal share of funding from public and private sources. Given the success of the initiated pre-breeding projects in apple, barley, perennial ryegrass, and plant phenotyping, NordGen and the governments of the Nordic countries have decided to continue funding this PPP.

Another successful PPP was established in 2012 in Southeast Asia by the International Potato Center (CIP), HZPC B.V. (a private Dutch potato seed company), and Syngenta Foundation for Sustainable Agriculture (SFSA) [37]. Whereas potatoes are usually regarded as a field crop, Drewnowski and Rehm [38] provided data that justify potatoes to be treated as a vegetable as well. This PPP aims at the collaborative breeding of five tropically adapted potato varieties with high and stable yields, thus enhancing the food security and family income of resource-poor farmers in Southeast Asia. Within a short period of only four years from the first crossings in 2016, five clones have already been identified for variety release in Vietnam. During the next phase, this successful PPP project aims at the development of processing varieties with multiple resistance against biotic stresses.

Phenotyping germplasm collections is laborious and costly, but international research initiatives and public-private partnerships have been established to mitigate this hurdle. For example, under the Horizon2020-funded G2P-SOL project (“Linking genetic resources, genomes and phenotypes of Solanaceous crops”), a collaboration of 19 institutions across Europe, Turkey, Israel, Peru, and Taiwan, global core collections of tomato, potato, pepper, and eggplant, have been generated, genotyped and phenotyped. The results of this endeavour are made publicly accessible [39].

The ECPGR coordinates the European Evaluation Network (EVA), involving genebanks, research institutes and private sector breeding companies and aiming to generate standardized evaluation data (both phenotypic and genotypic data) through participatory plant breeding actions [40]. Every partner contributes according to their expertise and capacity and especially the breeding companies implement field evaluations. The project evaluates accessions of wheat/barley, maize, carrot, lettuce, and pepper in crop-specific networks that currently bring together 29 participating genebanks from 21 countries, 49 breeding companies from 14 countries, and 34 research institutes from 20 countries for a 5-year period, from 2019 to 2024. It aims to establish a self-sustaining long-term project and to continue the evaluation of genebank accessions, also of additional crops. The evaluation activities would enable the participating breeding companies to observe and characterize new diversity that could potentially be of significant interest to them and would give them a few years of a leading edge over their competitors.

The International Lettuce Genomics Consortia (ILGC; [41] is another example of an efficient platform for the exchange of lettuce genetic diversity among breeders and scientists, worldwide. This consortium, led by UC Davis, USA, in which many countries participate and through which countries like the USA are actively distributing germplasm samples from their genebanks, including vegetable crops like lettuce.

Summarizing the different ways through which breeding companies acquire their genetic resources, it seems that the ‘traditional ways’ of collecting and acquiring from public genebanks are still ongoing. However, it is difficult to quantify at a global level the number of accessions and samples that are being obtained this way. A number of examples and cases are being presented, whenever possible, including data, to illustrate some common practices, including distribution figures from the Dutch and German national genebanks, from the WorldVeg genebank, including through PPP activities coordinated by them, and other examples of PPP projects from European countries as well as from the CGIAR centres. Through PPP initiatives, germplasm is exchanged in a very targeted manner. CGN provided details on joint collecting activities with private primarily Dutch breeding companies. They encountered limitations with the type of germplasm material available for use, in particular as many geographic areas are not accessible. Analysing the distribution data from CGN, IPK and WorldVeg, one could conclude that there is an ongoing flow of germplasm from the public genebanks to the breeding companies; in particular, the data from CGN show this clearly and those from WorldVeg illustrate how international agricultural research centres operate. At the same time, it should be noted that the information on the supply of germplasm materials to the private sector breeders is relatively scarce and limited, although some excellent examples of successful exchanges exist.

### 5.5. Benefit-Sharing Arrangements by the Private Sector

Whereas benefit-sharing aspects are not the main focus of this paper it was felt necessary to provide examples of concrete and/or perceived benefits shared with the countries of origin or directly with the farming communities that are regarded as the custodians of the genetic diversity and contributions made by the private sector to the global conservation efforts. It should be noted that especially the benefit-sharing with farmers and farming communities has become more complex and complicated due to the decision during the development process of the International Treaty to leave arrangements for Farmers’ Rights to the discretion of countries and not as a global responsibility. A second point that should be made in this context is the more recent and ongoing debate on digital sequence information (DSI), biological data associated with, or derived from, genetic resources such as nucleotide sequences and epigenetic, protein, and metabolite data. The benefit-sharing framework for DSI is currently being developed, based on a decision made by the Conference of the Parties of the CBD [42].

The following examples of benefit-sharing by the private sector have been mentioned or published and they demonstrate the different approaches and ideas that underly these arrangements:Support of Dutch companies to public national and local genebanks in building up, maintaining and regenerating collections as well as supporting collecting missions, thus contributing to long-term conservation;The establishment and operation of a regional (Afrisem) breeding programme by Rijk Zwaan and East-West Seeds in Tanzania allow contributions to the production and consumption of vegetables in Africa [43];Three global companies, as well as East-West Seed, reported collaborating with local partners to provide access to specific genetic material or biotechnology traits [43];Regional and national companies (e.g., East African Seed, Kenya Seed Company and Seed Co) work with partners in their country of origin, and partner with multiple local seedbanks and global research institutes by supporting genebanks and providing company genetic resources. Some national companies in East Africa also donate their germplasm to public research partners [43];Forty-four seed companies offer increasingly more extension services, including technical guidance and training to smallholder farmers in 47 countries on three continents [44];KWS reported the support of public genebanks in Peru and Ethiopia, and East-West Seed their support to genebanks in Indonesia and Thailand [43];Seven companies reported providing financial and technical support to the public (local/national) genebanks and four companies reported having given access to their own genetic resources [43].

These are just examples, and one may argue that these examples show that the extent of benefit-sharing is limited and incidental. Therefore, it is important to stress that the main (perceived) contribution of the breeding industry to farmers and growers is the added value that is comprised of the release of new varieties that they develop, combined with professional growing advice, that helps farmers achieve a better income. Through the open access system in plant breeding, i.e., the breeder’s exemption, these improved varieties are available to anybody for further breeding, including to farmers. Typically, such varieties possess new genetic diversity that allows better adaptation, increases yield and improves the nutritional value, even if the providers of the germplasm materials used in the breeding efforts are not necessarily the same as those that grow the new varieties.

## 6. Examples of Current Collaboration between Public Genebanks and Private Sector Breeders

Since many public genebanks have developed from germplasm working collections that had been established by predominantly public breeders, it could be expected that collaboration between the two is obvious and intense. However, with the growing importance of and attention to the conservation of threatened plant genetic resources, among others triggered by the leadership and coordinating role of the FAO and, to a lesser extent by the establishment of the Convention on Biological Diversity in 1992, countries had increasingly created national PGRFA programmes and built public genebanks. These developments resulted in more attention in the private sector to become attentive to genetic resources and to safeguard their own growing collections [3]. In addition, possibly stimulated by the global public and critical debate on ownership over PGRFA and on access and benefit sharing issues, the collaboration between the public research and conservation programmes, particularly in the main centres of crop origin and diversity, and breeding companies started to become more constrained. However, there are still some very good and convincing examples of a close collaboration between both sectors on the conservation and use of PGRFA and these are summarized below.

### 6.1. Centre for Genetic Resources (CGN)

Since its establishment, CGN has had a fruitful collaboration with the vivid breeding industry in the Netherlands. This was also built on the existing close collaboration between the pre-breeding programmes of the Dutch public breeding research institutes and the private breeding companies that had evolved over many decennia. CGN involves breeding companies (not only but predominantly Dutch) in many of its activities. The breeding companies advise CGN regarding technical issues, including the composition of the collections [45,46], and assist with the regeneration of the CGN accessions, as an in-kind contribution. Jointly with the companies, CGN also organises large-scale screening experiments of germplasm, among others in search of disease-resistant traits [47]. In these initiatives, the companies advise on what traits need to be traced and identified and on the respective screening protocol [48]. CGN distributes the germplasm accessions to the participating companies who screen them and send the results back to CGN, which combines the data, does a quality check (every accession is sent to two companies) and sends the combined data back to the respective participating companies. After an embargo period of usually three years, CGN makes all data publicly available through its online accessible database. The data are analysed and jointly published in a scientific paper (for example van Treuren et al. [48]). Besides the advisory role, the regeneration and joint phenotyping, companies are also involved in prioritising and funding collecting trips, including the benefit-sharing component, and other acquisition activities. In that context, they also provide material of their own varieties for inclusion in the CGN collection when these varieties are no longer on the market. Overall, CGN and the collaborating breeding companies have an intense and very positive collaboration, contributing to both conservation and use. Participation is, in principle, open for any company to join, initiatives are generally organised by CGN via Plantum, the Dutch association for plant breeders and young-plant growers.

### 6.2. World Vegetable Center (WorldVeg)

WorldVeg maintains the world’s largest vegetable genebank with 65,152 accessions encompassing germplasm of 133 genera and 330 species from 155 countries, including some of the world’s largest vegetable crop genepool collections held by a single institution, such as chilli pepper, tomato, and eggplant, as well as about 12,000 accessions of indigenous vegetables [49]. Due to a major regeneration backlog of primarily cross-pollinated vegetables, WorldVeg concluded agreements with private-sector companies to rescue original accessions in order to make them available to users worldwide. Companies willing to support WorldVeg in this endeavour include, among others Enza Zaden in the Netherlands for the rescue and multiplication of *Cucurbita moschata* (pumpkin) and *Momordica charantia* (bitter gourd) germplasm and Rijk Zwaan for the rescue and multiplication of *Citrullus lanatus* (watermelon) germplasm. Sakata Seed Corporation, Japan assisted the WorldVeg genebank with the screening of *Brassica* accessions for resistance to *Albugo macrospora* (white rust). The screening data are shared by WorldVeg online with the public after an embargo period of two years.

To accelerate the development and dissemination of elite vegetable crop materials, WorldVeg entered breeding consortia in Asia and Africa. The Asia and Pacific Seed Association (APSA)/World Vegetable Center Vegetable Breeding Consortium was founded in 2017 with 19 members and expanded to 51 in 2023 [50]. Participating companies obtain privileged early access to newly developed lines, for which they pay an annual fee, and are invited to an annual workshop to visit and evaluate field trials and interact with WorldVeg breeders. Feedback obtained from 34 vegetable seed companies in Asia that are part of this APSA/WorldVeg consortium showed that close to 90 commercial varieties of tomato, pepper, pumpkin, and bitter gourd that are currently sold in Asia contain pre-bred germplasm developed by WorldVeg [26]. APSA/WorldVeg consortium members sold 24.7 tons of seeds of these varieties in Asia in 2020. This quantity of seed is sufficient to plant vegetables on 171,000 hectares, benefitting close to half a million smallholder farmers in that region.

Given the positive response from the APSA/WorldVeg breeding consortium, a similar consortium was established in 2018 under the umbrella of the African Seed Trade Association (AFSTA). It is known as the Africa Vegetable Breeding Consortium (AVBC), which counted nine seed company members in 2019 and expanded to 23 members in 2021 [27,51]. AVBC membership grants early access to pre-bred material developed by WorldVeg breeders. During the 2021 AVBC workshop held in Arusha, Tanzania, private seed companies associated with AVBC, were able to evaluate 40 advanced breeding lines of African eggplant, amaranth, mungbean, peppers, pumpkin, and tomato [27].

### 6.3. East-West Seed International

The Genetic Resource Management section of East-West Seed International has its headquarters in Thailand and provides in-kind support to the national and other domestic genebanks in the Philippines and Indonesia. It also collaborates with CGN in regenerating germplasm materials [16]. Assistance is provided through regeneration support of accessions with low viability or low seed number, thus ensuring that these accessions are preserved for future generations.

In summary, this section describes the ongoing positive routine cooperation between the Dutch national genebank CGN and private sector breeding companies, largely from the Netherlands and coordinated by Plantum. The active engagement of several private companies in several routine operations of the genebank at WorldVeg as well as the active participation of breeding companies in breeding consortia in Asia and Africa using pre-bred materials from WorldVeg as parent lines demonstrate the advantages of such cooperation. The collaborative arrangements between (inter)national public genebanks and vegetable breeding companies, often coordinated by the respective seed associations, contribute significantly to germplasm collecting, conservation, documentation, and their sustainable use.

## 7. Opportunities for and Advantages of Collaboration between Breeding Companies and Public Genebanks

As already addressed above, the need to create genebank collections in breeding companies has been prompted by, among others, the decreasing access to ‘public PGR’. It can be observed, certainly in the Netherlands, that the willingness of the private sector to support public activities that contribute to increasing access to ‘public PGR’ is strong. There are various opportunities for further collaboration between the breeding companies and the public genebanks to create a win-win situation.

The provision of requested germplasm to breeding companies by genebanks serves as an obvious opportunity to explore, agree, and implement collaborative activities. Regarding the already mentioned germplasm exploration and collecting missions, it seems logical and advantageous to both, if the genebank would use its contacts and experience in planning and implementing collecting missions and, thus, potentially facilitate access to countries and regions that the breeding company otherwise would not have. Furthermore, it seems logical that the breeding companies would participate in the costs and if the collecting mission focuses strongly on the priority crops set by the breeding companies, the funding could be substantial. Such collaboration would also include the sharing of benefits with the countries in which germplasm is being collected.

Building on the specialized and deep knowledge of breeders of the crop(s) or crop genepool(s) they are focused on, it can be expected that the genebank could take advantage when structuring the collection into trait-orientated subsets, core collections, or other priorities and thus, increase the value of the conserved germplasm materials and increase their usability.

An obvious subsequent activity for the breeders to support public genebanks could be the multiplication of the collected materials of the crops of interest to the breeding company as they will have the knowledge and infrastructure to produce high-quality and healthy germplasm for subsequent long-term storage in the public as well as the in-house genebank. Such collaborations could also include germplasm materials already conserved in the public genebank, which require urgent regeneration. For example, from January 2012 to December 2019, more than 2100 accessions from the CGN genebank were regenerated and/or multiplied by private-sector seed companies. Most regenerations of CGN material are done by breeding companies, usually in various company locations in the Netherlands, and sometimes in other countries such as Spain or Morocco [1].

Another comparable opportunity is the joint molecular and/or phenotypic characterization of genebank accessions. Also, in this case all parties involved should benefit. Therefore, it can be attractive in those cases where the characterization is necessary for all parties in the consortium and implementing the characterization separately would only increase the costs. Implementing it jointly, coordinated by the genebank, and applying an embargo on the results for a couple of years after which the genebank can make the data public, can be a construction that would benefit all. CGN applies this approach in evaluating its collections for disease resistance as the participating breeding companies would all need to screen the germplasm for resistance anyway. It is the companies who decide the trait to be searched for and the method applied. It seems to be fair to conclude that the above-mentioned collaborative efforts would be mutually advantageous as the strengths of both partners are being combined. Such collaborations would also benefit other users of germplasm accessions in the country and worldwide as the knowledge on individual accessions will be steadily increased, and the quality of the germplasm samples will get to a higher standard.

Some public genebanks can offer specific expertise and knowledge on technologies such as cryopreservation, information technologies, seed science and molecular technologies that could be made available as part of a collaborative partnership with breeding companies. However, such opportunities are rare, as the breeding companies are generally better equipped. The area in which public genebanks do have a comparative advantage is their knowledge of where to obtain specific germplasm materials and/or related information and which genebanks might be accessible. Public genebanks might also be able to use their international reputation and credibility to facilitate access to germplasm, for instance as a contact point in organizing international collecting missions and/or exchanging germplasm.

Strengthened collaboration between public and private genebanks could significantly contribute to achieving a better complementarity between the strengths and interests of the two sectors, especially with respect to technology transfer, sharing of knowledge and expertise, sharing costs of joint activities, achieving more trust among the two sectors as a basis for more widely accepted legal and policy decisions. While considering such opportunities, one should clearly keep the motivations of both ‘sides’ in mind. The private sector is certainly willing to act in a ‘responsible way’ with respect to the genetic resources they obtained and possess, but they will rarely give up these strategic resources. At the same time, the companies need to ensure to have continued access to ‘new genetic diversity’ for the crops they breed. Thus, here lies possibly the most obvious area and common ground for both, the public genebanks in conserving genetic diversity and making it available for current and future use and the breeding companies in breeding new varieties on a commercial basis and thus directly contributing to food and nutrition security. The latter is also of greatest interest to the global, national, and local human societies, irrespective of whether ‘under development’ or ‘developed’ as long as just and transparent benefits at large are shared.

Given the above, the basic question is, whether we want to regard plant genetic resources as a public or a private resource. In this context, we equate the ‘common heritage’ principle of plant genetic resources with ‘a public good’, and this concept started changing during the last quarter of the past century. Although the ‘private good’ option seems tendency-wise to be decided, the authors believe that the arguments in favour of being a public good are strong and worth (re)considering. The breeding companies would certainly prefer that these genetic resources become a public resource (again), a ‘heritage of humankind’ style as initially treated by the International Undertaking. However, since the establishment of the CBD with the important notion that countries have the sovereign right over the genetic resources present in their territories and, consequently, regarding these resources as their property, private companies also need to treat genetic resources as a ‘commodity’. Considering this logic, we are sure that companies like to support developments that make the PGR more public again, albeit except for their own PGR. One must be aware and accept that the business of breeding companies is to breed, and that access to PGR is an essential prerequisite. Consequently, if society doesn’t give proper and facilitated access to genetic diversity, it could seriously hamper the breeding progress made by breeding companies. Therefore, private-sector companies must ensure that adequate genetic diversity is available to their breeders, enabling them to fulfil their mission to contribute to food and nutrition security.

## 8. Approaches to Facilitate Further Public-Private Collaboration

In this section, we identify and assess options for more public-private collaboration based on the discussions with breeders and based on the authors’ collective experience with hands-on public genebank and germplasm management practices over almost 50 years, on all continents of the world:Participation of private sector representatives in national genebank advisory committees. This seems to be a logical and important step that strengthens collaborations and thus, contributes to the sharing of responsibilities of joint interest. This would allow the voice of the private sector to be heard during the planning and implementation of the national genebank’s activities, allow for identifying and implementing complementary activities on ‘both sides’, formalize the collaboration, make it more visible and thus facilitating a better coordination and more efficient conservation and use at the national and maybe at the regional or even international level. Such a ‘formalization’ of the cooperation would undoubtedly increase the trust in each other and could eventually also facilitate discussions on contributions of the private sector to the implementation of the national PGRFA conservation strategy [52,53];Concluding formal, possibly long-term agreements between the public national genebank and breeding companies [53]. This will make the collaboration transparent and thus, facilitate/enhance the acceptability of the arrangement by the society at large, including politicians and thus, results in increased collaboration nationally, regionally, and globally. Furthermore, this will enable better planning of activities by all parties involved;Multilateral initiatives. These are in general the preferred format for collaboration [53], as they will increase the acceptability of the collaboration, ensure more sustainability, and combine a broader array of strengths and capacities to increase the sustainable use as well as the long-term conservation of the PGRFA that are part of such collaboration. One specific advantage is that such initiatives can involve regional partners that might share more common objectives. An excellent example of such an initiative is ECPGR which instigates and coordinates projects in facilitating the use of defined PGRFA through evaluation projects;A specific aspect of the previous point is the cooperation in broader consortia, e.g., EU-funded projects focusing on research activities, (re)sequencing genebank accessions; and joint participation in the ECPGR coordinated EVA project [40]. In general, this type of cooperation facilitates the generation of non-monetary benefits, such as the exchange of germplasm and information, access to and transfer of technology, and capacity building at a large(r) scale [5];Formalization of collaboration. An important aspect is the approval and support by the respective government(s) of the collaboration between the public genebank and the breeding company/ies [53] and thus, to strengthen also the sustainability of the governmental support to the public genebank This can be achieved by demonstrating the increased usefulness of the genebank for the society at large and the acquisition of additional funding for the genebank, for instance for collecting, characterization and evaluation activities enabled by the collaboration with the private sector;Provision of mutual services. For example, CGN organizes collecting missions; provides access to conserved germplasm materials and associated information; advises companies on use and legal (ABS) aspects and, where applicable, assists in the sharing of benefits. Dutch breeders, through their association with Plantum, provide policy advice to CGN through its crop advisory committees; provide technical inputs such as materials and knowledge through established working groups; support collecting missions, including co-funding and multiplication of collected materials; provide in-kind inputs, such as regenerating and evaluating accessions, morphological description as well as trait evaluation;Public genebanks and breeding companies should jointly look for opportunities to collaborate closely in convincing the society at large, including policymakers, that continued and unrestricted access to genetic resources will be the most efficient way to contribute to food and nutrition security. Furthermore, such collaboration will also generate ample benefits for all partners in the food value chain, to be shared with all these stakeholders in a just and transparent manner while recognizing the sovereign rights of states over their genetic resources and adhering to and/or achieving less bureaucratic and more user-friendly ABS regulations.

## 9. Conclusions and Recommendations

Whereas this paper has a focus on vegetable genetic resources and the sources of information are limited to a few countries and genebanks, the authors are convinced that the findings do apply to PGRFA at large, to their use in breeding activities in general and to public and private genebanks in many other countries and regions of the world.

Breeding companies started to establish their own genebanks at the beginning of the 1990s, based on their traditional working collections and later expanded these collections in a targeted way by acquiring wider genetic diversity of the crop genepools of their interest. It is argued that this development needs to be recognized in the planning and implementation of global and national conservation efforts and that cooperation between the public and private sectors is advisable to facilitate more sustainable, efficient, and effective collaborative efforts with regard to the long-term conservation and use of PGRFA.

Accepting the fact that access to PGRFA is becoming more restrictive and recognizing the need of breeders to have continued access to more and new genetic diversity, it will be indispensable to reform the current ABS arrangements as recently reviewed by Ebert et al. [5]. Such reform should duly consider the contributions the private sector makes to global conservation efforts, to the sharing of benefits (or lack thereof) as well as the importance of a continued flow of germplasm. Such a reform will be necessary to ensure the continued creation of new crop varieties that allow agriculture to cope with climate change and other constraints, thus contributing to more sustainable agriculture and global food and nutrition security.

The prevailing perception that the private sector is not contributing to the cost of long-term conservation of PGRFA undertaken by the public sector is not correct but makes it more difficult for countries to share their genetic resources freely and to arrive at an effective, inclusive, and rational oversight of any global PGRFA initiative and the global conservation and use system. Better and more targeted information, as outlined in this paper, on the role and contributions of the private sector to public conservation efforts is indispensable to change this perception.

The most fundamental reasons for the private sector to establish their own genebanks lay in the perceived shortcomings of the present legal framework, especially caused by the Nagoya Protocol, both for the ‘donors’ of the germplasm (i.e., predominantly the Global South) as well as the users (i.e., mainly private breeding companies in the North as well as in the Global South). These shortcomings are predominantly caused by the existing ABS arrangements, for which significant differences exist between expectations in the ‘Global South’ and the actual shared benefits by the ‘North’ to the biodiverse-rich countries, including the local custodians of this diversity. In this paper, we have taken the stance that the cooperation between private and public genebanks takes place within the existing legal framework of the ’global system’, especially regarding access and benefit-sharing aspects, particularly those of the International Treaty and its MLS. This is certainly the perspective that private breeding companies took while providing inputs to the paper in assessing reasons for establishing genebanks and in seeking improvements in collaboration with public genebanks. Consequently, the focus of this paper is on access to genetic resources, as the restrictions were the main reason for the private sector to establish genebanks and less on the benefit-sharing dimension. Nevertheless, and where possible, concrete (or perceived) examples of benefit-sharing contributions by the private sector have been presented in this paper. However, these examples might be seen as ‘fragmented’ and possibly ‘opportunistic’, but they give a more realistic and positive picture than what is usually experienced. It can be concluded that, despite these encouraging examples, much stronger incentives and a more facilitating legal ABS framework are needed to ensure that the (perceived) shortcomings can be resolved and thus, the collaboration between private and public genebanks can be further strengthened. The current ‘divide’ of countries with respect to their willingness to share PGRFA with others can only be overcome through improved communication making it clear to all that sharing of germplasm will be indispensable to achieve increased and long-lasting food and nutrition security globally. More transparent, less bureaucratic, and more efficient benefit-sharing arrangements are required to make this happen, at all levels.

It is argued that closer collaboration between private plant breeding companies and public sector genebanks in routine genebank operations, at the global, regional, and national levels, will benefit all, especially by strengthening the link between conservation and use [54]. This will lead to more efficient and targeted use of conserved genetic resources, to more cost-efficient conservation operations and thus, to a more sustainable agriculture. Increased trust, possibly achieved through better communication and accepting each other’s ‘business models’, will facilitate such cooperation.

It has become clear that an intensified cooperation between private plant breeding and public genetic resources conservation also requires a critical assessment of routine genebank operations, a more effective germplasm and information management, including improved ways of accession distribution that also facilitates the molecular use of the materials by the recipients.

## Figures and Tables

**Figure 3 plants-13-00247-f003:**
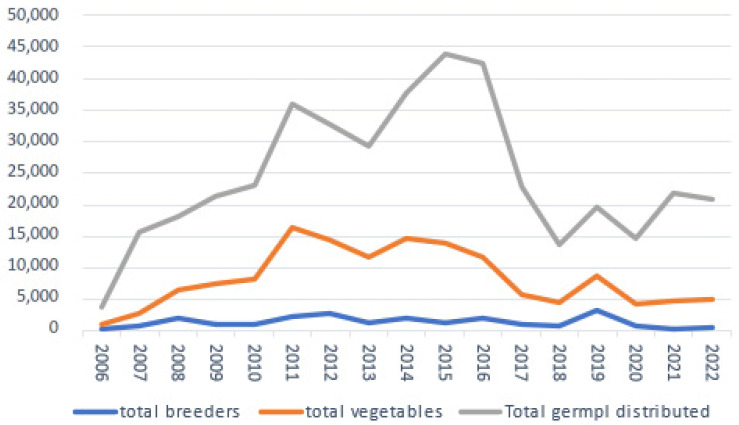
Numbers of distributed samples of vegetable crops by IPK to plant breeders only (blue line), to all users (orange), and total samples of all conserved crops and species distributed per year (light blue), from 2006–2022.

## Data Availability

No new data were created or analyzed in this study. Data sharing does not apply to this article.

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
