# Peer review of "Collaboration between Private and Public Genebanks in Conserving and Using Plant Genetic Resources"

_plants, 2024, doi:10.3390/plants13020247_

Round 1

Reviewer 1 Report

Comments and Suggestions for Authors

Author Response

Dear Reviewer,

Thank you very much for your critical review of our paper. We have appreciated your frank and clear comments on the draft and have attempted to address these points one by one carefully and comprehensively.

Your observation on the missing out of benefit-sharing aspects as part of our paper have been accepted with the understanding that the paper has been written with the understanding that access and benefit-sharing aspects have been implemented within the existing legal framework, especially of the International Treaty.

For details of our responses and revisions of the manuscript, please see the attachment.

With kind regards,

Johannes Engels, also on behalf of the two co-authors

Reviewer 2 Report

Comments and Suggestions for Authors

This manuscript discusses problems with the current system of plant germplasm exchange, especially between the public and private sectors, and proposes potential approaches to alleviate those problems. It serves the useful functions of describing the history of germplasm exchange, perceived problems with the current system from a private sector perspective, examples of successful collaborations, and recommendations to enhance collaboration. My major substantive comment is that the authors do not deal at all with digital sequence information (DSI), an increasingly relevant aspect of benefit-sharing of genetic resources. At a minimum the authors could acknowledge the importance and complexity of DSI and reference a recent article in Science magazine (Halewood et al., 2023, Science 382:520-522), or perhaps another article in this issue of Plants.

Specific comments and editing suggestions:

Figures 1 and 3; line 597-599 There appears to be a large difference between the Dutch and German genebanks in the percentage of samples sent to plant breeders/private sector. What are possible explanations for this?

Line 480-482. The percentages given here do not reflect what’s shown in Figure 3, e.g., for 2015, there are maybe 1500 samples sent to breeders out of nearly 45,000 total germplasm samples, far less than the stated 9.4%.

Line 488-490. Is there a reference or data to support this statement about the Indian, Chinese, and Russian genebanks?

Line 586. ‘bag’ should be ‘beg’.

Line 771-772. ‘online with the general public’ is stated twice.

Line 853. ‘in anyway’ is confusing and perhaps can be rephrased.

Line 854. Suggest replacing ‘to’ with ‘who will’.

Section 8. The points listed here would have greater impact and easier readability if the points were bulleted, enumerated, or italicized. Is the second point (‘Concluding formal, possibly long-term agreements …’) redundant with the fourth point (‘Formalization of collaboration’)?

Acronyms. The term ‘access and benefit-sharing’ and the acronym ABS are both used throughout the manuscript. After defining it for the first time in line 42, the acronym can be used. I wondered what PPP meant in line 400 and 639, then found it indirectly defined in line 640. In Figure 1, I suppose NLD means Netherlands, but it should be defined. ECPGR is used in line 644, but not defined until line 675. It doesn’t need to be defined again in line 930.

 In References, the page numbers for Byrne et al. are not stated correctly.

Author Response

Dear Reviewer,

Thank you for your critical reading and review of the manuscript. We have appreciated your comments and have responded to them in the attached file.

With thanks and regards,

Johannes Engels, also on behalf of the two co-authors
